# Analysis of social interactions and risk factors relevant to the spread of infectious diseases at hospitals and nursing homes

**Frederik Boe Hüttel**[1]*, **Anne-Mette Iversen**[2], **Marco Bo Hansen**[3], **Bjarne Kjær Ersbøll**[1], **Svend Ellermann-Eriksen**[4], **Niels Lundtorp Olsen**[1]

**1** DTU Compute, Technical University of Denmark, Lyngby, Denmark, **2** Department of Oncology, Aarhus University Hospital, Aarhus, Denmark, **3** Konduto ApS, Sani nudge, Copenhagen, Denmark, **4** Department of Clinical Microbiology, Aarhus University Hospital, Aarhus, Denmark

* fbohy@dtu.dk

**Data Availability Statement:** According to the Regional Ethics Committee (CEK) of the Capital Region of Denmark and the Danish Data Protection

## Abstract

Ensuring the safety of healthcare workers is vital to overcome the ongoing COVID-19 pandemic. We here present an analysis of the social interactions between the healthcare workers at hospitals and nursing homes. Using data from an automated hand hygiene system, we inferred social interactions between healthcare workers to identify transmission paths of infection in hospitals and nursing homes. A majority of social interactions occurred in medication rooms and kitchens emphasising that health-care workers should be especially aware of following the infection prevention guidelines in these places. Using epidemiology simulations of disease at the locations, we found no need to quarantine all healthcare workers at work with a contagious colleague. Only 14.1% and 24.2% of the health-care workers in the hospitals and nursing homes are potentially infected when we disregard hand sanitization and assume the disease is very infectious. Based on our simulations, we observe a 41% and 26% reduction in the number of infected healthcare workers at the hospital and nursing home, when we assume that hand sanitization reduces the spread by 20% from people to people and 99% from people to objects. The analysis and results presented here forms a basis for future research to explore the potential of a fully automated contact tracing systems.

## Introduction

During the ongoing COVID-19 pandemic, the safety of healthcare workers (HCWs) is of great importance to secure a functional level of staffing at hospitals and nursing homes. HCWs are at high risk of SARS-CoV-2 exposure through direct or indirect contact with infected patients, colleagues or equipment [1, 2]. With the upsurge in hospital admissions, this pandemic is threatening to leave some healthcare systems overstretched and unable to operate effectively [3].

One effective method to reduce the risk of transmission is effective and timely contact tracing [4–7] which allows for containment of the pathogen by isolating potentially infected

Agency, the raw sensor data cannot be made publicly available because the data contains tracking of employees at hospitals and nursing homes. The tags contain time and place stamps that, in combination with the other information (e.g., a rotation schedule), allow for potential identification of the healthcare workers. We are allowed to share the aggregated data presented in the article but not the raw data. The contact person in the Regional Ethics Committee (CEK) of the Capital Region of Denmark is Margit Sonne Bom, cand. jur. (detik@detik.dk) and the contact person in the Data Protection Agency is Line K. Sørensen, cand. jur. (dt@datatilsynet.dk).

**Funding:** BKE - 1 grant by Innovation Fund Denmark (J. no. 0208-00045B). The fund did not play any role in the study design, data collection and analysis. The fund also did not decide to publish or prepare the manusciprt.

**Competing interests:** The authors have declared that no competing interests exist.

individuals. When an HCW is diagnosed with COVID-19, the healthcare organisation is responsible for carrying out the contact tracing and identifying close contacts of the infected. Currently, contact tracing is a manual process which makes it time-consuming, resource-heavy and slow. If an HCW is tested positive for COVID-19, the person must remember interactions with colleagues to identify who has been exposed and should self-isolate. Recalling every social interaction is hard which in turn makes contact tracing very imprecise. Recent research has highlighted these issues with traditional contact tracing and argue for data-driven methods to make contact tracing efficient [8, 9]. With the onset of the SARS-CoV-2 pandemic, there has been an increased focus on the importance of proper hand hygiene to reduce the spread of diseases. Inspired by this, we here analyse data from an automated hand hygiene system [10, 11], focusing on the interactions between HCWs. The main objective of this study is to investigate the social interactions that occur at hospitals and nursing homes and identify risk factors relevant to the spread of SARS-CoV-2. We aimed to use the data from the automated hand hygiene system. We will use Monte-Carlo methods to estimate the number of infected HCWs and how many of them should be quarantined. The main contributions of this study are:

- Evaluation of the risk of spread of infectious disease among HCWs.

- Identification of rooms which are central for the spread of disease in which HCWs should be especially aware to wear personal protective equipment, sanitize hands and clean surfaces.

- Estimation of the numbers of infected HCWs in case of infectious disease outbreak.

This study is part of a larger research collaboration between Technical University of Denmark, Konduto, Aarhus University Hospital, and Sølund nursing home. The research project aims to develop automated data-driven contact tracing, based on the hand hygiene monitoring system. One of the benefits from this would be that only HCWs at risk are required to quarantine and to be tested when there is an outbreak.

## Methods and material

### Study design and setting

We conducted a prospective, observational study between August 13, 2020, and November 11, 2020, in a Danish university hospital (Aarhus University Hospital, 4 wards, 64 beds) and a nursing home facility (Sølund Nursing Home, 6 floors, 156 apartments).

**Study subjects.**   In the hospital, the study subjects included nurses (n = 123), doctors (n = 86), and cleaning staff (n = 11). In the nursing home, the study subjects included both nurses and nurse assistants combined (n = 64). Participation of the study subjects was voluntary. We did not monitor HCWs who did not participate. Five HCWs from both the hospital and nursing home did not participate. Data were anonymised to both study participants and investigators. The HCWs in the nursing homes were stratified into the following groups: day shifts (7.30 am to 3.30 pm), evening shifts (3.30 pm to 11.30 pm) and night shifts (11.30 pm to 7.30 am).

**Data collection.**   Data were collected using an automated hand hygiene system (Sani nudge, Copenhagen, Denmark, https://saninudge.com. Accessed Aug 5, 2021).

**System setup.**   We here provide a brief description of the automated hand hygiene system. Bluetooth sensors were placed on existing alcohol-based hand rub dispensers and above patient beds. Anonymous Bluetooth sensor IDs were placed on the existing name badges of HCWs and encoded with the occupation of the ID wearer. Furthermore, the sensors on the

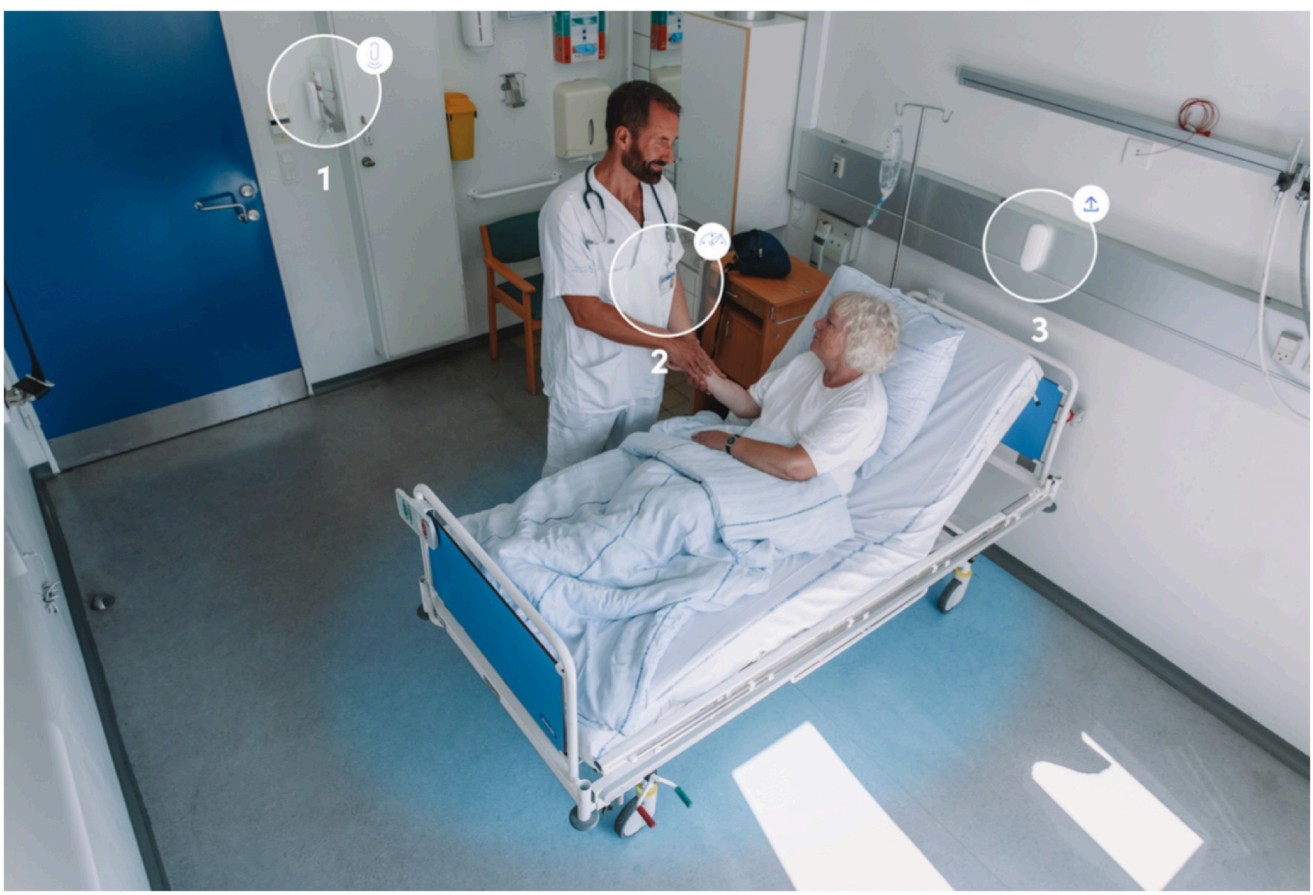

**Fig 1. The Sani nudge hygiene system in a hospital patient room. 1**. A Sensor on a dispenser registering the number of alcohol-based hand rubbing events; **2**. An anonymous sensor on the name badge of each healthcare worker. The sensor is coded according to staff profession. **3**. The patient zone (illustrated with blue) created by the sensor near the head of the patient bed. Data captured by the sensors are sent to a secure cloud-based server and stored at the device level. The two persons in the figure are volunteers and not part of the study.

hand rub dispensers register when the dispenser is being used and nudges nearby HCWs to perform hand hygiene using discrete lights [10, 11].

The stationary sensors continuously tracked the presence of nearby Bluetooth sensor IDs, indicating the presence of an HCW. This data is combined into a *hand hygiene event*, giving the location of an HCW (anonymized and known only by its sensorID) and whether the HCW performed hand hygiene according to the WHO guidelines for patient interaction [10]. The sensors were placed in rooms where hand hygiene is important, such as patient rooms and medication rooms at the hospital and apartments (resident homes) and kitchens at the nursing home A complete list of rooms can be found in S1 File. The system has been clinically validated [12]. Fig 1 contains an overview of the system setup in a patient room.

**Encounters.**   A single hand hygiene event has a location, time, duration and ID. The data locates HCWs into specific rooms for a specific time-frame. If the system detected two events within the same room at the same time-window, we infer that two HCWs have met at that location. We define this meeting or interaction between two individuals as an *encounter* between the two HCWs. These encounters naturally define the edges of a social network.

**Social network.**   A social network describes the social interactions between members of a community. Graph theory and social networks have a rich tradition in epidemiological research going back to the 1980s [13, 14], and have been used to perform contact tracing in case of an infectious disease outbreak [15–18]. Through network analysis, we can identify which person has been in contact with an infected person or identify chains of encounters from a person to an infected person. We can use the network structure to identify central members, which plays a central role in the spread of an infectious disease.

## Modelling

We consider the set of nodes $V$ and the set of edges $E$, which defines a social graph $G = (V, E)$. We include both rooms and HCWs in the set of nodes $V$, and $E$ contains the encounters inferred between two nodes in $V$. We use the set of edges to identify paths of transmission of an infectious disease. We define two different type of edges in the network.

1. Edges between two different HCWs

2. Edges from the two HCWs to the room, where the encounter occurs.

We include edges between persons and rooms, because a virus could be transmitted to surfaces in these rooms and further transmit the disease to subsequent HCW using the rooms.

**Centrality measure.**   One of the advantages of using a social network in infectious disease analysis is the ability through centrality measures to identify key nodes in the network [18]. The centrality of a node $v$, indicates the ability for that node to transmit the infectious disease to other nodes. The degree centrality of a node $v$ is the fraction of nodes in $V$ it is connected to. The degree centrality is normalised by dividing by the maximum possible degrees of a node, which is the length of V minus 1, as self-loops are excluded:

$$C_D(v) = \frac{\deg(v)}{|V| - 1} \tag{1}$$

**Dynamic graph.**   Not only is the encounter between HCWs important for the spread of disease, but the time of the encounter is vital, too. For an edge between two HCW to be able to spread a disease, one of the HCW must have been infected or infectious at the time of the encounter. We extend the social graph to include the temporal dynamics of the disease spread. We still keep the entire set $V$, however, denoting the temporal edges $E_t$, where t is the time-step in the simulation. The temporal dynamics allow the edges to evolve, as new social interactions occur, and old ones are updated based on social interactions. An example of a dynamic graph can be seen in Fig 2.

Using the social graph we can analyse individuals who might be infected by analysing chains of infected HCWs. We can also use the dynamics of the network to determine when an HCW was possibly infected.

## Simulations

For assessing the number of infected people by a disease outbreak, we used a Monte-Carlo based approach using the SIR-model [19] dynamics on the social graph. The SIR model was adopted due to its common use in epidemiological research [15, 19, 20]. In a SIR model, individuals in a population transition between the states of *Susceptible*, *Infectious* and *Recovered* (SIR), similar to the approach in [20]. We model the state of employees as a binary boolean

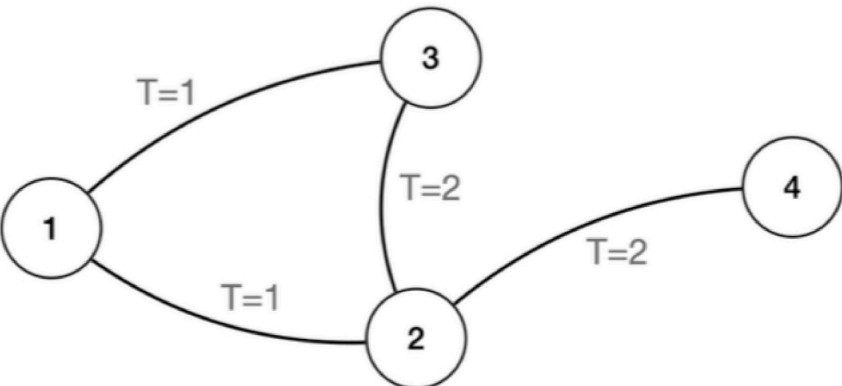

**Fig 2. Here a dynamical graph is depicted.** There are 4 nodes (1,2,3 and 4), which represent 4 different HCWs. At $t = 1$, node 1,2 and 3 connect; then at $t = 2$, node 2, 3 and 4 connect.

vector $Y_t$ containing all nodes in the set of nodes

$$y_{i,t} = \begin{cases} 1 & \text{if node } i \text{ is infected} \\ 0 & \text{otherwise} \end{cases} \tag{2}$$

The state indicates the state of a node either Infectious (1) or Susceptible (0). Assume we have an employee $y_{i,t}$ who is not sick at time $t$, and has an encounter with employee $y_{j,t}$ for a duration of $t'$, we then compute the probability of $y_{j,t}$ being sick based on the duration of the encounter and the state of $y_{i,t}$. We use this probability as the weight of the edge $e_{\{i,j\},t}$ in the set $E_t$.

**Reed-Frost.** We model the epidemic using a Reed-Frost model [21], which is a discrete-time epidemic model. Initially, at time-step $t = 0$ we let a randomly chosen person be infectious and the rest of the network be susceptible. Individuals who are affected at time $t$ infect susceptible connected nodes in $G$ by edge $e_t$ independently with probability $p$. Those who become infectious at time $t$ are the infectious $t + 1$ and can spread the disease through $E_{t+1}$. Even though the true transmission probability $p$ is unknown, some kind of Poisson transmission dynamics is often assumed [15, 22–24]. Here we model the probability of transmission $p$ as an independent Poisson process with a rate $\lambda$ and time interval as the observed time in encounter $t'$.

$$e_{\{i,j\},t} = p(y_j = 1 \mid y_j = 0, y_i) = \begin{cases} 0, & y_i = 0 \\ 1 - e^{-\lambda t'} & y_i = 1 \end{cases} \tag{3}$$

Using the estimated transmission probabilities (Eq 3), we set up a stochastic simulation. When 2 HCWs meet we draw Bernoulli random variables using the estimated $p$ for the distribution to simulate if an HCW transmits the disease to another HCW. We also note that if a node is susceptible and not sick, we will allow for any spread of disease across the edge connecting the two nodes. We assume that any transmission of diseases between individuals follows the probability from Eq 3. The rate $\lambda$ can be altered to model different diseases or the same disease with or without sufficient personal protective equipment. The $\lambda$ is the number of events we expect within a 15-minute time window. We applied three different $\lambda$ values in our simulations:

- $\lambda = 1$, which is equal to a high probability of being infected if a contact has been 15 minutes or longer.

- $\lambda = 15$, which means that there is a high probability of spreading the disease after each minute of an encounter

- $\lambda = 200$, which is an extreme case. Where there is a high probability of transmission even for encounters lasting a few seconds. We use this to identify a worst-case scenario in case of a disease outbreak.

Using these three different values we evaluate a range of scenarios, from the extreme cases to the more realistic scenario.

**Hand sanitizer.** The effect of using hand sanitizer in respiratory diseases transmission between individuals is difficult to determine in the general society [25–27]. Previous studies have estimated the effect of using hand sanitizer to be within 16–21% reduction in transmission in the general public [25–27], and for transmission to rooms (surfaces and equipment) to be a reduction of 99% [28]. In these simulations, we assume that if an HCW uses hand sanitizer, we reduce the transmission probability $p$ by either 20% for HCW to HCW and 99% for HCW to a room. We opted for a 20% reduction which is a round value within the interval.

We weight the edges in the network according to transmission probabilities computed using Eq 3. If the system observes a hand sanitation then the probability is reduced by 20%. We assume that HCWs can spread the disease for 4 days, before they contract symptoms and stay at home. The 4 day threshold was selected to account for the variability (between 2 and 4 days) in the presymptomatic spread of SARS-CoV-19 [29].

## Ethics

Pursuant to the Danish law, approval was queried and evaluated as not needed by both the Ethics Committee (J. no. 20028629) and the Danish Data Protection Agency (J. no. 2020–211-4867). Participation in the study was voluntary. Participants were informed verbally and received written information about the project. Informed consent was given by the participants active choice to pickup and carry a tag during their working hours.

## Results

In total, the system observed 89791 hand hygiene opportunities from the hospital wards and 18590 hand hygiene opportunities from the nursing home facility, which resulted in 17008 encounters (HCW interactions) at the hospital and 4717 encounters at the nursing home.

The average meeting duration between two individuals was higher for nursing homes (191 seconds) compared to the average meeting duration at the hospitals (100 seconds). We found that hospital meetings are frequent and short, compared to the nursing home where the encounters are less frequent but with a longer duration. Only a small fraction (Hospital = 1.1% and Nursing homes = 5.3%) of the actual meetings occur for a length of more than 15 minutes (Table 1). Health authorities use a threshold of 15 minutes of exposure to an infected person to determine if an HCW should self-quarantine. We observe that the risk of being exposed to an infected person for more than 15 minutes is not likely. We see that the risk of 15 minutes continued exposure is higher at nursing homes compared with hospitals.

**Table 1. Overview of the duration of the different observations at the different locations.** Not accumulated.

| Location | 0–1 minut | 1–5 minute | 5–10 minutes | 10–15 minutes | > 15 minutes |
|---|---|---|---|---|---|
| Nursing Home | 33.7% | 39.5% | 15.8% | 5.6% | 5.3% |
| Hospital (all rooms) | 46.7% | 43.0% | 7.3% | 1.8% | 1.1% |
| Hospital (only patient rooms) | 46.2% | 49.1% | 4.2% | 0.4% | 0.001% |

## Staff distributions

The nurses accounted for the majority of the hand hygiene opportunities and interactions that took place between the HCWs in the hospital (Fig 3). The nurses spend the most time in the different rooms, whereas the doctors rarely spent more than 5 minutes inside a room (Fig 3).

At the nursing home, we observe a similar spread between the different employee types (Fig 4). The night guards have a slightly lower number of observations compared to the day guard and the afternoon guard. The distribution of meetings appears to be similar across the different HCWs types (Fig 4). The employees at the nursing home have comparable tasks to complete during a working day, and there is not a large difference between the day shift and afternoon shift. We find that the distribution of HCW types are spread equally at the nursing home, compared to the hospital, where the nurses account for a large majority.

## Centrality measure

We here present the centrality measures (Eq 1) for the different nodes at the two locations. We used a dynamical graph that spans four days. This duration reflects the duration where an HCW is assumed to be infectious to the time when they start to show symptoms. Once the HCW shows symptoms, the HCW must self-isolate and notify management, to initiate contact

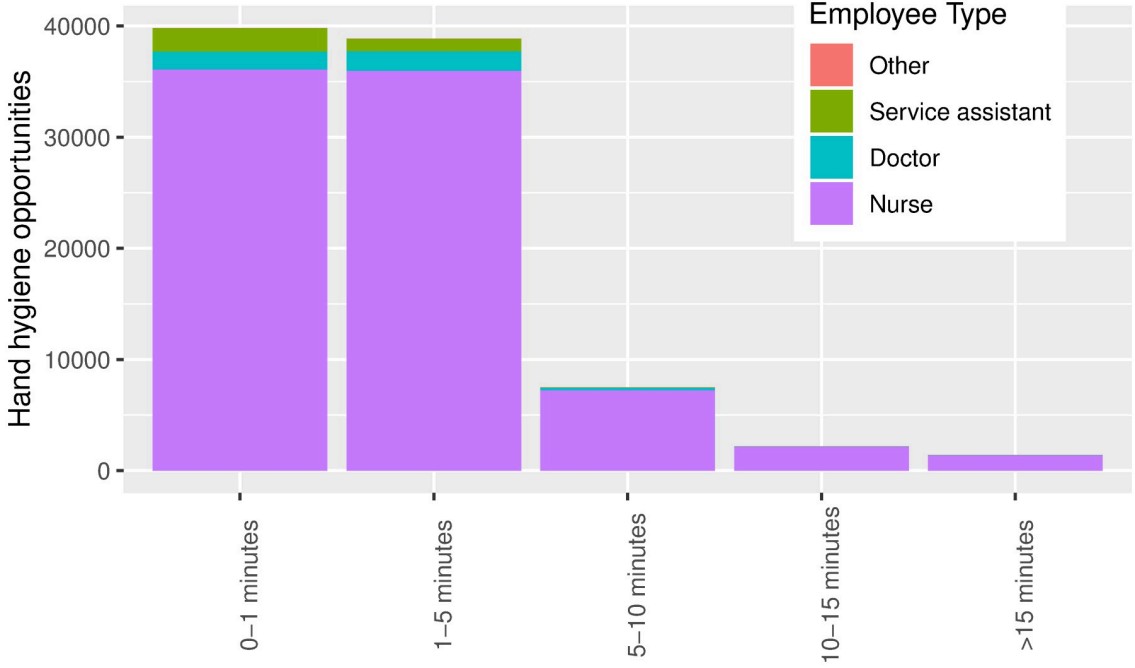

**Fig 3. Distribution of encounter durations of the different types of HCW at Aarhus University Hospital.**

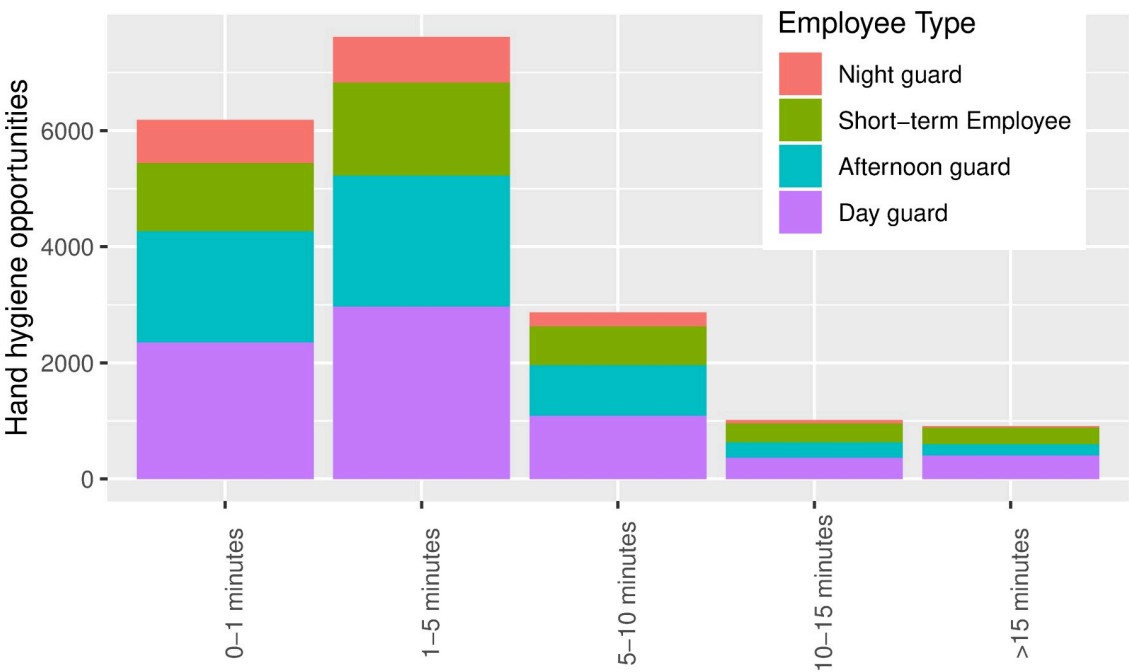

**Fig 4. Distribution of encounter duration of the different types of HCW at Sølund nursing home.**

tracing. We used a period of four days to assess the risk of spread at the locations. The simulation uses a time-step of 15 minutes.

In the hospital, we found that medication rooms are by far the most central locations in the social network. This indicates there is an increased risk of disease transmission in medication rooms because many HCWs have edges connecting to this room (Fig 5). We also see that the nurses are central nodes in the network because they have many connections when compared to the centrality measure of a doctor.

At the nursing home, we find an equal distribution of the centrality measure compared to the hospital (Fig 6). The kitchen is the place where most employees are present at the same time, which indicates that this is an important risk area. Also, we observe interactions occurring in the hallway. At the nursing home, the hallway is where residents spend time with leisure activities and social gatherings. In the hallway, HCWs should be cautious, as they interact with residents but also with other HCWs.

Common for both locations is that we found that the rooms where there exists an interaction between an HCW and a patient/resident are not the most central rooms. However, we find that rooms, such as the kitchens and the medication rooms, play far more important roles in the structure of the social networks.

## Results from simulations

We used Eq 3 with different λ to simulate different scenarios. We adjusted λ to simulate different levels of infectiousness and HCWs use of masks and other protective gear. In the social network, we identify 123 nodes at the hospital (43 rooms and 80 HCWs) and 52 nodes at the Nursing home (29 HCWs and 23 rooms). We found that for the extreme case (λ = 200) there is no need to isolate and quarantine the entire staff and clean every room (Table 2.). For the more reasonable cases (λ = 1 and λ = 15) we see that only a few rooms and HCWs get infected, and therefore a complete shutdown is not required.

## Boxplot of the centrality measure for the different type of nodes

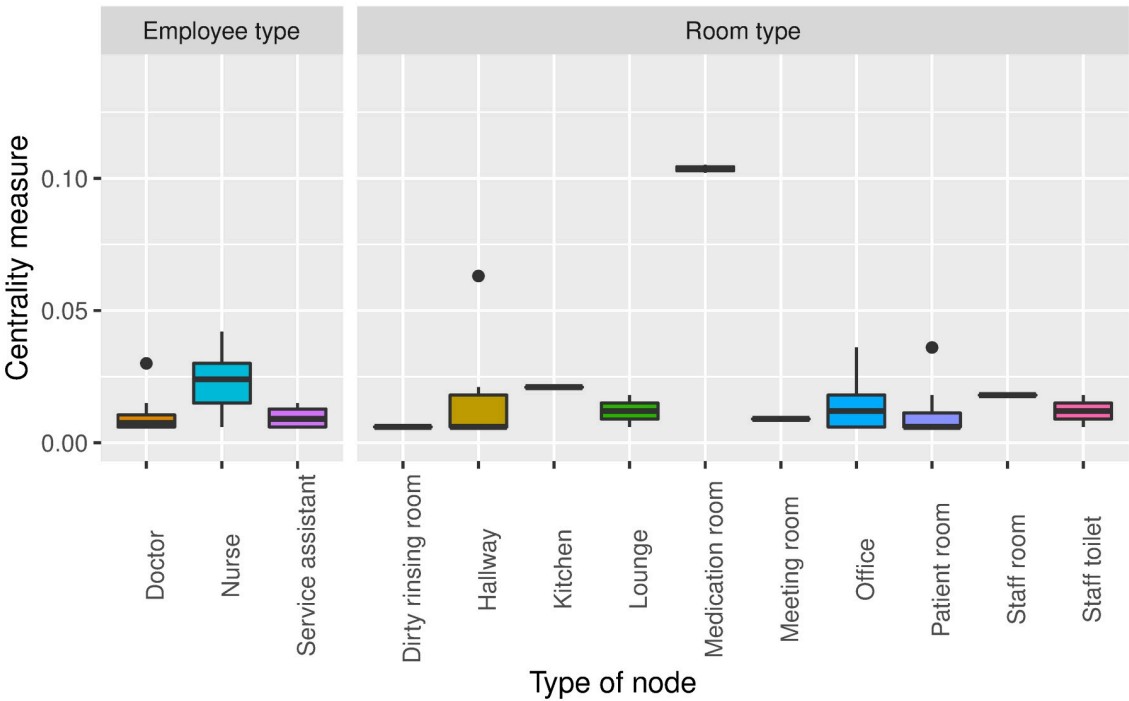

**Fig 5. Boxplot showing rooms where the duration of the encounters are above 15 minutes at Aarhus University Hospital.**

## Boxplot of the centrality measure for the different type of nodes

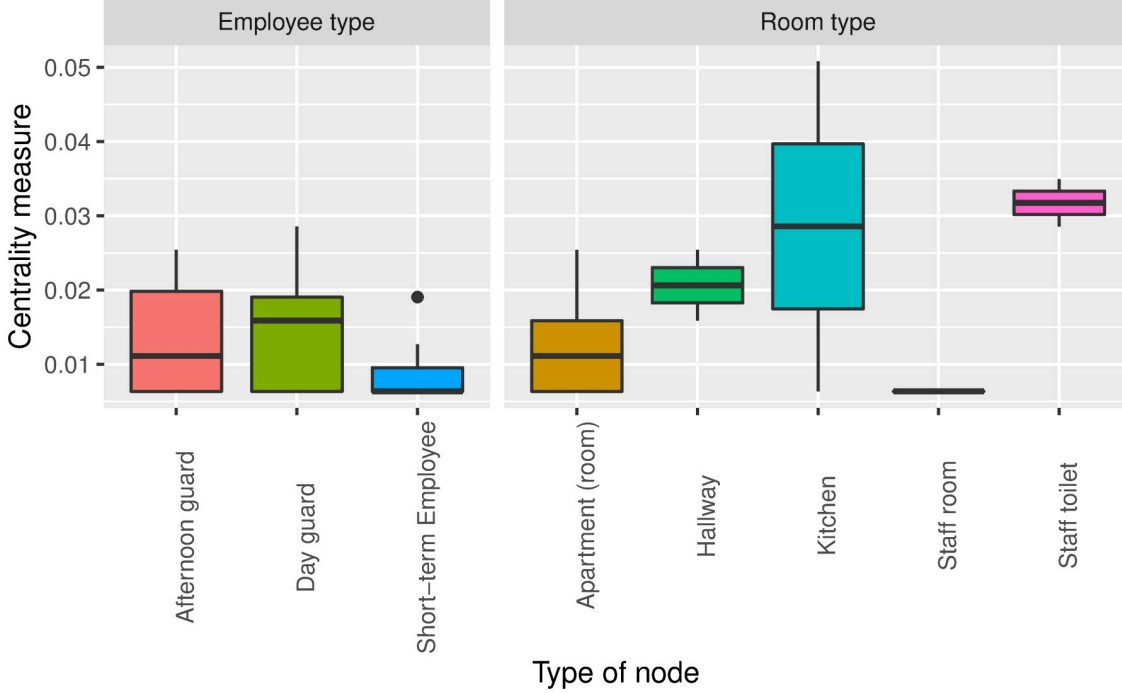

**Fig 6. Boxplot showing rooms where the duration of the encounters are above 15 minutes at Sølund nursing home.**

Table 2. Number of infected rooms and HCWs at the different locations using different λ in Eq 3.

| Location | λ = 1 | λ = 15 | λ = 200 | λ = 15 with sanitizer |
|---|---|---|---|---|
| Nursing Home | 2.14 | 12.57 | 30.17 | 9.20 |
| Hospital | 2.46 | 17.31 | 63.31 | 10.27 |

For both locations (Table 2, Figs 7 and 8), we observe a reduction in the number of infected people, when we included hand sanitizing in the models. At the hospital, we see a lowering on an average of 7.04 infected HCWs and at the nursing home 3.37 HCWs using λ = 15, which corresponds to a 41% and 26% reduction in infected HCWs. Even if we underestimate the effect of hand sanitizer, there appears to be a reduction in the number of infected persons.

## Discussion

Through the analysis and results, we have identified rooms which are central to the spread of disease at hospitals and nursing homes. These rooms are areas of risk where HCWs should be extra careful to follow the infection prevention guidelines, such as social distancing, wearing masks and sanitizing hands.

Interestingly, we found that a majority of encounters in the hospital (98.9%) and the nursing home (94.5%) are shorter than 15 minutes which otherwise is the time-period used as a guideline for being classified as a close contact in Denmark [30]. To our knowledge, the 15 minutes cut-off is based on theoretical assumptions and consensus and is not evidence-based, as for COVID-19 cases are coming from multiple shorter exposure to infected [31]. Our results question the clinical relevance of this cut-off time as it appears conservative. We find that a

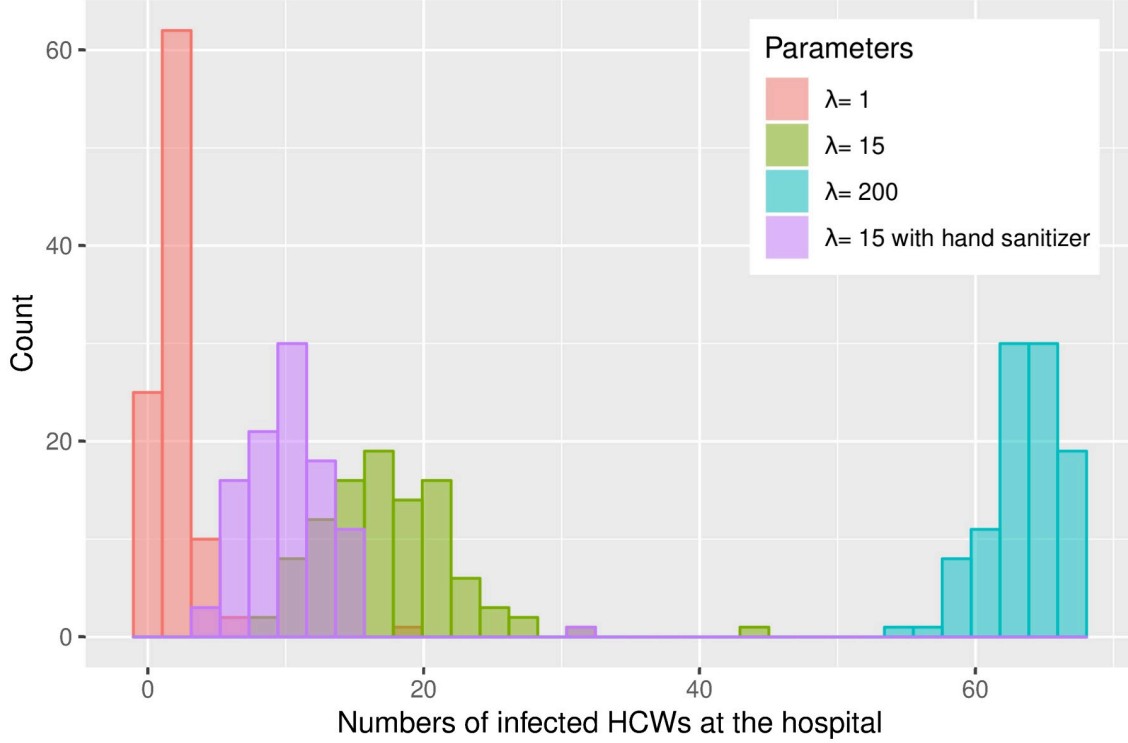

Fig 7. Histogram over the number of infected and the frequency over 100 simulations at the hospital.

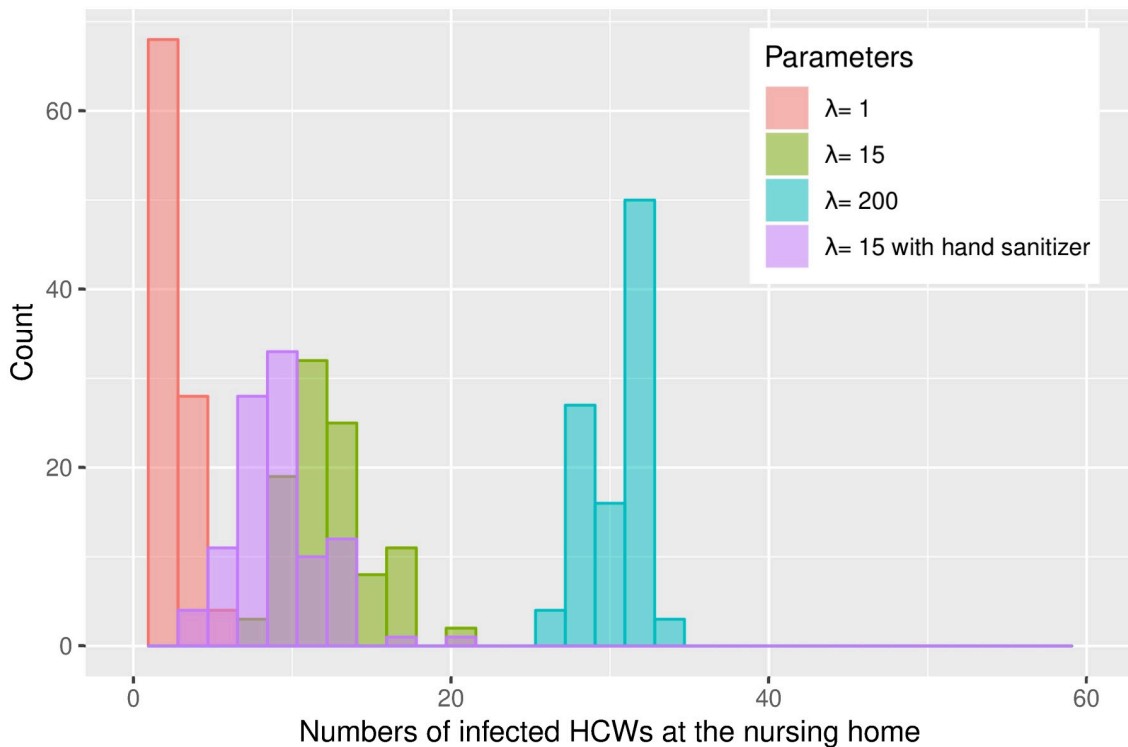

**Fig 8. Histogram over the number of infected and the frequency over 100 simulations at the nursing home.**

contact tracing system must account for short encounters down to encounters between 1 to 5 minutes to identify a majority of contacts to be effective and work in clinical practice, and use this to calculate the cumulative exposure time to an infected. When relying on manual contact tracing, short encounters can be difficult to remember which introduces a high risk of recall bias and potentially forgetting important interactions with an infected person. One of the strengths of an automated contact tracing system is that it can identify these short encounters and use it to identify and alert exposed personnel who should quarantine. This can be done quickly and effectively which minimises the risk of further disease transmission as otherwise seen with time-delaying processes during manual contact tracing. We therefore argue that the research presented here provides a basis for future research to explore a fully automated contact tracing system.

We recognise some of the limitations of our approach. First, we only monitored HCWs but it might also be relevant to collect data on visitors once the healthcare organisations allow visitors again. In nursing homes, the residents spend much of their time in common areas where interactions with other residents take place. These interactions constitute paths for disease transmission but were not the focus of this study. Likewise spread of disease from hospitalised patients to the staff has not been studied here. Given the current circumstances at nursing homes and hospitals wards without COVID-patients, it is more likely that HCWs contract the disease outside of the workplace, and bring the virus to work where spread of disease takes place [32].

Some HCWs might have altered some of their routines, knowing that they are being monitored. Previous research has shown that the HCWs alter their routines when they are nudged using light to use hand sanitizer more, which in turn increases the sanitation of the locations [10, 11].

Second, we did not install the sensors in the offices, nursing stations, and kitchens where some of the social interactions take place. These interactions are therefore not included in the analysis. The locations of the sensors were carefully selected by their importance on hand hygiene according to official guidelines and by extension, their relevance for disease spread. However, we believe that interactions in the offices and kitchens also pose a risk of disease transmission, and should be included in future studies. We remark that we used a simplified model of the rooms, and do not account for HCWs wearing Infection Control and Prevention (ICP) equipment. We did not incorporate and account for cleaning of the rooms, as rooms no longer are infectious after being cleaned. However, we argue that for the case where a patient is infected, cleaning the patient room will not prevent the disease from spreading from patient to HCW. To account for this, we let the rooms follow the same dynamics as HCWs in the simulations. Ideally, knowledge of which rooms contain a patient can alleviate this limitation.

Third, these simulation-based results have not been clinically validated during disease outbreaks in the wards. Ideally, the transmission probabilities would be estimated using statistical models. These statistical models would provide a basis for assessing the actual probability of an HCW contracting an infectious disease. With further analysis of the data and the statistical models, the effect of hand sanitizer to reduce the spread of respiratory disease could be measured.

Based on the points made in this research paper, we believe there is justification for future work to expand on this social network structure by including all room types and by investigating the probability of transmission between HCWs, which we here assumed to be determined by an independent Poisson process.

## Conclusion

In conclusion, we have analysed the social interactions and risk factors which are relevant to the spread of infectious diseases at hospitals and nursing homes. We generated a dynamical social graph using an already installed hand hygiene monitoring system. Through analysis of the social graph, we identified medication rooms and kitchens as key rooms where HCWs should be particularly aware of following the infection prevention guidelines as there is a high risk of infectious disease transmission due to a high number social interactions occurring in these rooms. We observed a reduction of 41% and 26% of the number of infected HCWs, from the hospital and nursing home, respectively, when we account for the use of hand sanitizer. When considering the extreme case of a very contagious disease and no hand sanitation, we find that 14.1% and 24.2% of HCWs in the hospitals and nursing homes are potentially infected.

## Supporting information

**S1 File. Locations of sensors.** Complete list of locations where sensors where placed. (PDF)

**S2 File. Current practice of contact tracing.** Brief description of practices for contact tracing at the time of publication. (PDF)

## Acknowledgments

We thank all the HCWs who took part in this study.

## Author Contributions

**Conceptualization:** Bjarne Kjær Ersbøll.

**Data curation:** Frederik Boe Hüttel, Anne-Mette Iversen.

**Formal analysis:** Frederik Boe Hüttel.

**Funding acquisition:** Marco Bo Hansen, Bjarne Kjær Ersbøll.

**Investigation:** Frederik Boe Hüttel, Anne-Mette Iversen.

**Methodology:** Frederik Boe Hüttel, Bjarne Kjær Ersbøll, Niels Lundtorp Olsen.

**Project administration:** Bjarne Kjær Ersbøll, Niels Lundtorp Olsen.

**Resources:** Marco Bo Hansen.

**Supervision:** Svend Ellermann-Eriksen, Niels Lundtorp Olsen.

**Validation:** Frederik Boe Hüttel, Anne-Mette Iversen, Marco Bo Hansen.

**Visualization:** Frederik Boe Hüttel.

**Writing – original draft:** Frederik Boe Hüttel, Anne-Mette Iversen, Marco Bo Hansen.

**Writing – review & editing:** Frederik Boe Hüttel, Anne-Mette Iversen, Marco Bo Hansen, Bjarne Kjær Ersbøll, Svend Ellermann-Eriksen, Niels Lundtorp Olsen.

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
