## [Decision Letter · Decision Letter 0]

8 Jul 2021

PONE-D-21-07190

Automated contact tracing to prevent the spread of coronavirus in hospitals and nursing homes

PLOS ONE

Dear Dr. Hüttel,

Thank you for submitting your manuscript to PLOS ONE. After careful consideration, we feel that it has merit but does not fully meet PLOS ONE’s publication criteria as it currently stands. Therefore, we invite you to submit a revised version of the manuscript that addresses the points raised during the review process.

We look forward to receiving your revised manuscript.

Kind regards,

Florian Fischer

Academic Editor

PLOS ONE

Journal Requirements:

2. 1) Is is possible for others to obtain access to the same dataset? If so, please provide details of this in the Data availability statement. 2) Did the authors have access to any identifying information, or was the dataset anonymised when received?

3.We note that you have indicated that data from this study are available upon request. PLOS only allows data to be available upon request if there are legal or ethical restrictions on sharing data publicly. For information on unacceptable data access restrictions, please see http://journals.plos.org/plosone/s/data-availability#loc-unacceptable-data-access-restrictions.

5. Please ensure that you refer to Figure 2 in your text as, if accepted, production will need this reference to link the reader to the figure.

6.We note that Figure 1 includes an image of a patient / participant in the study. 

Reviewers' comments:

Reviewer's Responses to Questions

**Comments to the Author**

1. Is the manuscript technically sound, and do the data support the conclusions?

Reviewer #1: Yes

Reviewer #2: Partly

2. Has the statistical analysis been performed appropriately and rigorously? 

Reviewer #1: I Don't Know

Reviewer #2: I Don't Know

3. Have the authors made all data underlying the findings in their manuscript fully available?

Reviewer #1: No

Reviewer #2: No

4. Is the manuscript presented in an intelligible fashion and written in standard English?

Reviewer #1: Yes

Reviewer #2: Yes

5. Review Comments to the Author

Reviewer #1: This is an interesting study. Given the differences between the Danish and US healthcare systems, I do have some questions and comments. Why did you note the different shifts of HCW in the nursing home and not the hospital? Do the nurses in the hospitals work rotating shifts? In the text you mention that you monitored nurses and nursing assistants in the nursing homes, but the graph shows guards and not nurses or nursing assistants. Please correct the text to match the graph. Lots of hospitals in the US do not use medication rooms to a great extent now since meds are sent from the pharmacy to the nursing units individually labeled for patients. Thus nurses do not have to count out and dispense medications, unless they are prn for pain. Why did you not include nursing stations as places of common interactions? All units still have a central nursing station and that is where nurses are between patient care. Is the lounge mentioned the same as a staff break room where lunches occur? I know that hand sanitizer outlets are located in halls and all patient rooms, but are they also in areas where no patient care occurs?

Reviewer #2: The manuscript addresses a mainstream topic, and its results may be used to improve the overall approach to positive COVID-19 cases in the healthcare workforce, therefore proving to be useful to clinical and public health practice. It is written in cautious language and provides data on social interactions outside the commonly studied workplaces. The analytical part of the methods is developed to fully detail and in a fluent manner. There is no doubt concerning the pertinence and importance of its findings, which deserve to be published as they contribute to expand our scientific knowledge on the area, with practical implications in day-to-day practice.

It claims that social interactions between healthcare workers - and between workers and rooms - occur almost exclusively below the 15-minute threshold considered by the WHO; in fact, most interactions are shorter than 5 minutes. These interactions happened mainly in the medication room, but authors also claim that passing-through areas are not negligible and need to be taken into consideration. It also claims, using mathematical simulations, that for each putatively infected worker, only 2 to 3 other nodes – either workers or rooms – may be infected, provided preventive measures are applied and the virus behaves as currently expected. Even in a worst-case scenario, there would be no need to isolate every worker and clean every room, effectively shutting down the premises. The authors go on to claim that their analysis highlighted some limitations of manual contact tracing that could be solved using an automated system.

The paper has two main strengths. It uses an implemented system for detection of interactions based on Bluetooth sensors located in key areas. This allows researchers to identify every node-to-node interaction without facing recall bias, thus improving the reliability of the data collected. This system is comprehensive enough to include areas such as the hallway, the kitchen and staff toilet, which are usually absent in studies considering the same issue, and its great sensitivity make it tough for participants to manipulate the observers. The paper also differentiates interactions between the two most important settings in healthcare in terms of SARS-CoV-2 transmission – hospitals and nursing homes – which contrast widely in terms of social interactions, healthcare management, organization and purpose.

The manuscript has 2 major areas that require an improvement, which are: the overall connection between title, abstract, objectives and conclusion; and the comprehensiveness of the discussion.

Regarding the first, the authors write that the main objective was “(…) to investigate the social interactions that occur at hospitals and nursing homes and identify risk factors relevant to the spread of SARS-CoV-2.” (lines 38-39). This has no direct link with the title. While automated contact tracing may draw on the conclusions of this manuscript, it is not the topic being directly studied. The title should reflect the main finding of the paper, in line with its main objective, and that finding concerns social interactions. The same is true for the conclusions. The argument on how an automated system could improve manual tracing by better identification of interactions would be well placed in the discussion, but goes beyond the scope of the conclusions. I would recommend the conclusion to focus on key rooms of interactions, as it does, and on simulations’ results, if relevant. On the other hand, I found no risk factors being directly identified through simulations. Therefore, I would suggest the second objective to adapt better to the analysis performed.

Regarding the discussion, the authors consider both rooms and workers in the of nodes V, as both may be sources of transmission (line 101). While this is true, the dynamics of transmission from room to worker or from worker to worker are different. In their simulations, for λ=1, only a few nodes get infected, slightly above 2 for each setting. In practice, whether the node is a room or a worker has markedly different implications – rooms are cleaned and become immediately operational, while workers need to be isolated for a certain time period, which affects normal healthcare delivery. These issues are not present in the discussion, which I find to be underdeveloped given the tremendous potential these findings have to discuss disease dynamics, health policy and public health interventions.

There are other areas requiring improvement, though smaller adjustments are necessary, that I report as they appear in the submission. These are minor issues, that nevertheless need to be addressed.

In the abstract, there is a claim for a” (…) 41% and 26% reduction in the number of infected healthcare workers at the hospital and nursing home.” These values are not present throughout the article, even though the overall simulations are presented. I would advise for consistency between abstract results and manuscript results.

Lines 42-47, on contributions of the study, would be better placed on the discussion section.

Line 53: the authors claim that, in case of an outbreak, an entire section of workers need to be quarantined. Does this not go against the idea of contact tracing, where only high-risk contacts are isolated, as explained in lines 31-32?

Line 64: Participation of subjects is voluntary. There should be some comparison made between participants and non-participants, to understand whether the sample may be representative of the population or not. This is not even addressed in the discussion.

Line 129: No explanation is given for the choice of the SIR-model for this infection. A small sentence would suffice.

Line 170: While authors give estimations of previous studies, the rationale behind choosing 20% as the transmission probability for worker to worker is not given. Even if it may be because, within the interval 16-21%, 20% may be the easiest to analyze, that should be made clear.

Line 174: No explanation for the assumption that workers may spread the disease for 4 days. Pre-symptomatic infectious period is usually considered to be 2 days, though at least one paper found a 3-day mean. Some reference is needed.

Lines 218-226: Observing figure 6, why is the staff toilet not addressed in this section?

Line 244: Overestimated appears to have been written instead of underestimated.

Line 267-onwards: No sentence discussing the potential observation bias, as workers may have altered some of their routines based on being observed. This may even link with their will to participate,, affecting the study's representativeness. Even if authors regard it as negligible, as it may well be, it still should be discussed.

Line 276: “Second, we did not install the sensors in all the offices and kitchens where some of the social interactions take place”. The supporting information provided does not clarify which kitchens and offices have senses, and whether they differ from other kitchens and offices or not. It is unclear how relevant this is – authors claim sensors were carefully located. I believe this could be further developed, and some consideration on representativeness should be present, as observed for worker’s participation.

Line 286: Sentence ends without verb, it seems as if it ends in the middle.

Figure 5 and 6 have the exact same label, though addressing different settings.

6. PLOS authors have the option to publish the peer review history of their article (what does this mean?). If published, this will include your full peer review and any attached files.

Reviewer #1: No

Reviewer #2: No

---

## [Author Response · Author response to Decision Letter 0]

22 Aug 2021

Reply Letter

Response to reviews

We thank the editor and the reviewers for evaluating our manuscript and are pleased to hear that our manuscript is considered for publication in Plos One. Below is our response to each point and comment raised by the editor and reviewers. 

Sincerely,

Frederik Boe Hüttel, PhD Student, Technical University of Denmark

Journal Requirements:

Author response: File naming was edited to comply with the style requirements. We hopefully have no divergences from the style requirements now. 

2. 1) Is it possible for others to obtain access to the same dataset? If so, please provide details of this in the Data availability statement. 

2) Did the authors have access to any identifying information, or was the dataset anonymised when received?

Author response: Though data in the dataset are pseudonymised, data are legally treated as personal data and thus subject to GDPR. 

3.We note that you have indicated that data from this study are available upon request. PLOS only allows data to be available upon request if there are legal or ethical restrictions on sharing data publicly. For information on unacceptable data access restrictions, please see http://journals.plos.org/plosone/s/data-availability#loc-unacceptable-data-access-restrictions.

Author response: According to the Regional Ethics Committee (CEK) of the Capital Region of Denmark and the Danish Data Protection Agency, the raw sensor data cannot be made publicly available because the data contains tracking of employees at hospitals and nursing homes. The tags contain time and place stamps that, in combination with the other information (e.g., a rotation schedule), allow for potential identification of the healthcare workers. We are allowed to share the aggregated data presented in the article but not the raw data. The contact person in the Regional Ethics Committee (CEK) of the Capital Region of Denmark is Margit Sonne Bom, cand. jur. (detik@detik.dk) and the contact person in the Data Protection Agency is Line K. Sørensen, cand. jur. (dt@datatilsynet.dk).

Author response: Please refer to the preceding answer.

Author response; ORCID ID for the corresponding has been added and validated in Editorial Manager to comply with the requirements. 

5. Please ensure that you refer to Figure 2 in your text as, if accepted, production will need this reference to link the reader to the figure.

Author response: We have added references to the figure.

6.We note that Figure 1 includes an image of a patient / participant in the study. 

Author response: The two persons seen in the figure are volunteers (not patients and not healthcare workers) to illustrate how the system works to comply with the data processing law in our country. They have signed a standard consent form which we can provide to you upon request. 

Reviewer 1:

1) This is an interesting study. Given the differences between the Danish and US healthcare systems, I do have some questions and comments. 

Why did you note the different shifts of HCW in the nursing home and not the hospital? Do the nurses in the hospitals work rotating shifts? 

Author response: Thank you for the valuable comments. It was decided by the two healthcare organizations how they wanted the data to be collected, grouped and presented. The nursing preferred the data to be stratified according to the shifts, whereas the hospital chose the staff profession. It is simply a matter of what data grouping they felt confident with and thought made most clinical value when presented to the HCWs.

In the text you mention that you monitored nurses and nursing assistants in the nursing homes, but the graph shows guards and not nurses or nursing assistants. Please correct the text to match the graph. 

Author response: Thanks for the comment. It has now been corrected. 

Lots of hospitals in the US do not use medication rooms to a great extent now since meds are sent from the pharmacy to the nursing units individually labeled for patients. Thus nurses do not have to count out and dispense medications, unless they are prn for pain. Why did you not include nursing stations as places of common interactions? All units still have a central nursing station and that is where nurses are between patient care. Is the lounge mentioned the same as a staff break room where lunches occur? I know that hand sanitizer outlets are located in halls and all patient rooms, but are they also in areas where no patient care occurs?

Author response: This is a valid point. We agree that the nursing stations are places of frequent interaction. However, we do not have nursing stations in Denmark, but it corresponds to ‘offices’ in the manuscript (page 10, line 286), included in the limitation section. In this study, we did not install the system in these room types, but it could be interesting to see the results of future studies where these room types are included. We have added ‘nursing stations’ to the limitation section to accommodate international readers and avoid misunderstandings. 

Reviewer 2:

The manuscript addresses a mainstream topic, and its results may be used to improve the overall approach to positive COVID-19 cases in the healthcare workforce, therefore proving to be useful to clinical and public health practice. It is written in cautious language and provides data on social interactions outside the commonly studied workplaces. The analytical part of the methods is developed to fully detail and in a fluent manner. There is no doubt concerning the pertinence and importance of its findings, which deserve to be published as they contribute to expand our scientific knowledge on the area, with practical implications in day-to-day practice.

Author response: Thank you for the comments.

It claims that social interactions between healthcare workers - and between workers and rooms - occur almost exclusively below the 15-minute threshold considered by the WHO; in fact, most interactions are shorter than 5 minutes. These interactions happened mainly in the medication room, but authors also claim that passing-through areas are not negligible and need to be taken into consideration. It also claims, using mathematical simulations, that for each putatively infected worker, only 2 to 3 other nodes – either workers or rooms – may be infected, provided preventive measures are applied and the virus behaves as currently expected. Even in a worst-case scenario, there would be no need to isolate every worker and clean every room, effectively shutting down the premises. The authors go on to claim that their analysis highlighted some limitations of manual contact tracing that could be solved using an automated system.

The paper has two main strengths. It uses an implemented system for detection of interactions based on Bluetooth sensors located in key areas. This allows researchers to identify every node-to-node interaction without facing recall bias, thus improving the reliability of the data collected. This system is comprehensive enough to include areas such as the hallway, the kitchen and staff toilet, which are usually absent in studies considering the same issue, and its great sensitivity make it tough for participants to manipulate the observers. The paper also differentiates interactions between the two most important settings in healthcare in terms of SARS-CoV-2 transmission – hospitals and nursing homes – which contrast widely in terms of social interactions, healthcare management, organization and purpose.

The manuscript has 2 major areas that require an improvement, which are: the overall connection between title, abstract, objectives and conclusion; and the comprehensiveness of the discussion.

Author response: Thank you for pointing this out. We have now incorporated your suggested changes below, and we believe that it has 

improved the manuscript considerably. Please see our answers below for detailed explanation about the changes.

Regarding the first, the authors write that the main objective was “(…) to investigate the social interactions that occur at hospitals and nursing homes and identify risk factors relevant to the spread of SARS-CoV-2.” (lines 38-39). This has no direct link with the title. While automated contact tracing may draw on the conclusions of this manuscript, it is not the topic being directly studied. The title should reflect the main finding of the paper, in line with its main objective, and that finding concerns social interactions. The same is true for the conclusions. The argument on how an automated system could improve manual tracing by better identification of interactions would be well placed in the discussion, but goes beyond the scope of the conclusions. I would recommend the conclusion to focus on key rooms of interactions, as it does, and on simulations’ results, if relevant. On the other hand, I found no risk factors being directly identified through simulations. Therefore, I would suggest the second objective to adapt better to the analysis performed.

Author response: We have changed the title to better reflect the main findings of the paper and reworded parts of the conclusion, focusing on the findings of interactions in key rooms and the simulation results. 

Regarding the discussion, the authors consider both rooms and workers in the of nodes V, as both may be sources of transmission (line 101). While this is true, the dynamics of transmission from room to worker or from worker to worker are different. In their simulations, for λ=1, only a few nodes get infected, slightly above 2 for each setting. In practice, whether the node is a room or a worker has markedly different implications – rooms are cleaned and become immediately operational, while workers need to be isolated for a certain time period, which affects normal healthcare delivery. These issues are not present in the discussion, which I find to be underdeveloped given the tremendous potential these findings have to discuss disease dynamics, health policy and public health interventions.

Author response: We are aware of some of the simplifications of the rooms, and we have few sentences to discuss it, which is lines 293-298. However, we argue that for the case where a patient is infected, cleaning the patient room will not prevent the disease from spreading from patient to HCW. To account for this, we let the rooms follow the same dynamics as HCWs in the simulations. Ideally, knowledge of which rooms contain a patient can alleviate this limitation.

There are other areas requiring improvement, though smaller adjustments are necessary, that I report as they appear in the submission. These are minor issues that nevertheless need to be addressed.

Author response: Thanks for highlighting these points, which we have addressed point-by-point below.

In the abstract, there is a claim for a” (…) 41% and 26% reduction in the number of infected healthcare workers at the hospital and nursing home.” These values are not present throughout the article, even though the overall simulations are presented. I would advise for consistency between abstract results and manuscript results.

Author response: The percentages were extracted from Table 2, and have been included in the text and the conclusion now.

Lines 42-47, on contributions of the study, would be better placed on the discussion section.

Author response: We agree with the comments made by the reviewer that contributions of a study are also well placed in the discussion section. However, we believe the contributions placement in the introduction helps the reader contextualise our paper with respect to our contribution. Therefore, we have kept the contributions in the introduction section. If the reviewer has a strong opinion about this, we will move it to the discussion section. 

Line 53: the authors claim that, in case of an outbreak, an entire section of workers need to be quarantined. Does this not go against the idea of contact tracing, where only high-risk contacts are isolated, as explained in lines 31-32?

Author response: Thanks for the comment. We have corrected the part.

Line 64: Participation of subjects is voluntary. There should be some comparison made between participants and non-participants, to understand whether the sample may be representative of the population or not. This is not even addressed in the discussion.

Author response: Almost all HCWs at the two locations participated in the study and due to the nature of the study, we cannot monitor HCWs who did not participate. HCW, who participated in the study, picked up the sensors themselves from a box placed in a recreational room. We estimate that a majority of the HCWs participated from both locations.

Line 129: No explanation is given for the choice of the SIR-model for this infection. A small sentence would suffice.

Author response: We have added a small sentence to model choice.

Line 170: While authors give estimations of previous studies, the rationale behind choosing 20% as the transmission probability for worker to worker is not given. Even if it may be because, within the interval 16-21%, 20% may be the easiest to analyze, that should be made clear.

Author response: We have added a small explanation to the 20%.

Line 174: No explanation for the assumption that workers may spread the disease for 4 days. Pre-symptomatic infectious period is usually considered to be 2 days, though at least one paper found a 3-day mean. Some reference is needed.

Author response: We have included references and arguments to why 4 days have been chosen.

Lines 218-226: Observing figure 6, why is the staff toilet not addressed in this section?

Author response: We believe the other points mentioned are more important for the analysis and discussion.

Line 244: Overestimated appears to have been written instead of underestimated.

Author response: Thanks for spotting the spelling mistake.

Line 267-onwards: No sentence discussing the potential observation bias, as workers may have altered some of their routines based on being observed. This may even link with their will to participate, affecting the study's representativeness. Even if authors regard it as negligible, as it may well be, it still should be discussed.

Author response: Previous research into hand sanitisation has evaluated the effect (Hawthorne effect). We have added a sentence on the matter, this is in lines 282-286. Importantly, previously studies using the same automated hand hygiene system as in this paper have shown in comparison analyses that HCWs quickly fall back into old routines and habits (approx. after two weeks). We accounted for this in this study using a 2 months baseline period (control period) to limit the risk of the observer effect. As other studies have investigated this specifically, and it was not part of this study aim, we hope that the reviewer finds the added sentence on this matter to be sufficiently addressed. 

Line 276: “Second, we did not install the sensors in all the offices and kitchens where some of the social interactions take place”. The supporting information provided does not clarify which kitchens and offices have senses, and whether they differ from other kitchens and offices or not. It is unclear how relevant this is – authors claim sensors were carefully located. I believe this could be further developed, and some consideration on representativeness should be present, as observed for worker’s participation.

Author response: No kitchens and offices had sensors installed. We have deleted “all” to avoid misunderstandings. The sensors were placed based on guidance from the infection prevention and control guidelines and team. However, we do believe that future studies should include these room types to account for all interactions. We have addressed this in line 286-291. 

Line 286: Sentence ends without verb, it seems as if it ends in the middle.

Author response: We have completed the sentence.

Figure 5 and 6 have the exact same label, though addressing different settings.

Author response: We have changed the caption to reflect the different settings.

---

## [Decision Letter · Decision Letter 1]

8 Sep 2021

Analysis of social interactions and risk factors relevant to the spread of infectious diseases at hospitals and nursing homes

PONE-D-21-07190R1

Dear Dr. Hüttel,

We’re pleased to inform you that your manuscript has been judged scientifically suitable for publication and will be formally accepted for publication once it meets all outstanding technical requirements.

Kind regards,

Florian Fischer

Academic Editor

PLOS ONE

Additional Editor Comments (optional):

Reviewers' comments:

Reviewer's Responses to Questions

**Comments to the Author**

1. If the authors have adequately addressed your comments raised in a previous round of review and you feel that this manuscript is now acceptable for publication, you may indicate that here to bypass the “Comments to the Author” section, enter your conflict of interest statement in the “Confidential to Editor” section, and submit your "Accept" recommendation.

Reviewer #2: All comments have been addressed

2. Is the manuscript technically sound, and do the data support the conclusions?

Reviewer #2: Yes

3. Has the statistical analysis been performed appropriately and rigorously? 

Reviewer #2: I Don't Know

4. Have the authors made all data underlying the findings in their manuscript fully available?

Reviewer #2: (No Response)

5. Is the manuscript presented in an intelligible fashion and written in standard English?

Reviewer #2: Yes

6. Review Comments to the Author

Reviewer #2: The authors have addressed all corrections and suggestions. Title, objectives and conclusion are now coherent between them. Discussion is only slightly improved, yet that was the choice of the authors.

7. PLOS authors have the option to publish the peer review history of their article (what does this mean?). If published, this will include your full peer review and any attached files.

Reviewer #2: No

---

## [Editor Report · Acceptance letter]

10 Sep 2021

PONE-D-21-07190R1 

Analysis of social interactions and risk factors relevant to the spread of infectious diseases at hospitals and nursing homes 

Dear Dr. Hüttel:

I'm pleased to inform you that your manuscript has been deemed suitable for publication in PLOS ONE. Congratulations! Your manuscript is now with our production department. 

Kind regards, 

on behalf of

Dr. Florian Fischer 

Academic Editor

PLOS ONE